# ADELT: Transpilation Between Deep Learning Frameworks

## Abstract

We propose the **Ad**versarial **DE**ep **L**earning **T**ranspiler (**ADELT**), a novel approach to source-to-source transpilation between deep learning frameworks. ADELT uniquely decouples code skeleton transpilation and API keyword mapping. For code transpilation, it uses few-shot prompting on large language models, while for API keyword mapping, it employs contextual embeddings from a code-specific BERT. These embeddings are trained in a domain-adversarial setup to generate a keyword translation dictionary. ADELT is trained on an unlabeled web-crawled deep learning corpus, eschewing hand-crafted rules and parallel data. It outperforms state-of-the-art transpilers, improving exact match scores by 17.4 pts and 12.0 pts for PyTorch-Keras and PyTorch-MXNet transpilation pairs respectively. We provide open access to our code, corpus, and evaluation benchmarks.

## 1 Introduction

The rapid development of deep learning (DL) has led to an equally fast emergence of new software frameworks for training neural networks. Unfortunately, maintaining a deep learning framework and keeping it up-to-date is not an easy task. Many deep learning frameworks are deprecated or lose popularity every year, and porting deep learning code from a legacy framework to a new one is a tedious and error-prone task. A *source-to-source transpiler between DL frameworks* would greatly help practitioners overcome this difficulty.

Two promising solutions to source-to-source transpilation between deep learning frameworks are unsupervised neural machine translation (NMT) (Artetxe et al., 2018) and large language models. NMT treats deep learning code as a sentence for training sequence-to-sequence (Sutskever et al., 2014) models, but its applicability is limited due to the scarcity of parallel corpora and its notable data hunger. On the other hand, large language models like GPT-3 (Brown et al., 2020), pretrained on web crawl data, offer potential, performing translation tasks in a few-shot or zero-shot manner. Our early experiments with Codex (Chen et al., 2021), a GPT-3 model specialized for code, show its potential in few-shot transpilation of deep learning programs. Yet, such models struggle with API-specific nuances, inaccurately handling function names and parameter mappings. These limitations underscore the difficulties large language models face in preserving precision in complex real-world applications.

That said, most deep learning framework code is *structured*: each type of layers has its own constructor, and constructing a network involves calling each layer's constructor in a chaining manner. By leveraging the structures of programming languages, we can *decouple* the transpilation of skeletal codes from the mapping of API keywords. The transpilation of skeletal codes is the easier part, and large LMs already do a great job. We only need a separate algorithm to translate the *API keywords*, i.e., the function and parameter names to complete the transpilation.

In this paper, we present ADELT (Figure 1), a method that leverages this insight to transpile DL code. ADELT outperforms the state-of-the-art end-to-end transpilers. The canonicalized source code is decoupled into two parts: the code skeleton and the API keywords. ADELT transpiles the code skeleton using a pretrained large language model by few-shot prompting. Each API keyword occurrence is then embedded into a vector by PyBERT, a BERT pretrained on Python code. This vector is both the textual and the contextual representation of the API keyword. ADELT then leverages domain-adversarial training to learn a generator that maps the vector to an aligned embedding space. The alignment is enforced by a two-player game, where a discriminator is trained to distinguish

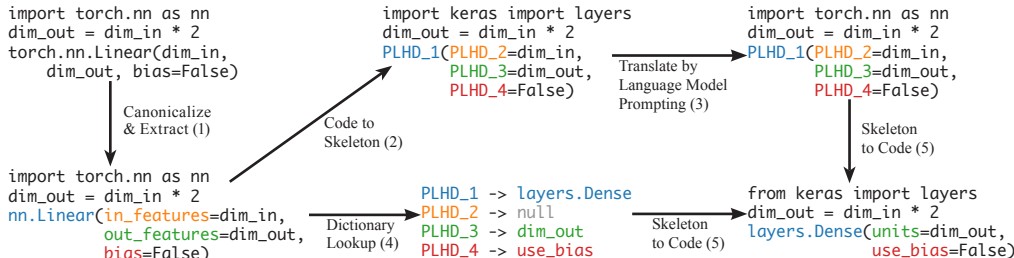

Figure 1: **An example of ADELT's pipeline:** an import statement in the code skeleton is transpiled from PyTorch to Keras by a language model via few-shot prompting; a linear fully-connected layer is transpiled by removing the argument `in_features` and renaming other API keywords according to the learned dictionary. The number (1 to 5) near each arrow label corresponds to the step number in Section 2.

between the embeddings from the source DL framework and those from the target DL framework. The API keyword embeddings are trained jointly with the generator as the output embedding matrix of a softmax classifier on the aligned embedding space. After generating a synthetic API keyword dictionary from the embeddings using a two-step greedy algorithm, ADELT then looks up each API keyword occurrence in the dictionary and puts them back into the transpiled code skeleton.

In summary, this paper makes the following contributions:

- We introduce ADELT, a robust solution for transpilation between deep learning frameworks without training on any labeled data. Outperforming large language models, ADELT excels across various transpilation pairs, achieving exact match scores of 73.0 and 71.5 for PyTorch-Keras and Keras-PyTorch transpilations, respectively. These scores surpass those of the state-of-the-art large language model for code, GPT-4, by 17.4 and 6.8 points respectively.
- To demonstrate our technique, we construct a PyTorch-Keras-MXNet corpus of deep learning code from various Internet sources, containing 19,796 PyTorch modules, 3,703 Keras layers/models, and 1,783 MXNet layers/models. We then build an evaluation benchmark for PyTorch-Keras and PyTorch-MXNet transpilation. The benchmark evaluates both our API keyword mapping algorithm and the overall source-to-source transpilation.

## 2 METHOD

**ADELT** (**A**dversarial **DE**ep **L**earning **T**ranspiler) is an algorithm that transpiles code from a source deep learning framework into an equivalent one in a target framework, by transpiling the skeletal code using a pretrained large language model, and then looking up each keyword in a dictionary learned with unsupervised domain-adversarial training. ADELT applies the following steps to each piece of input code, which we illustrate using the example shown in Figure 1:

1. Extract *API calls* from the source code. Such API calls can be automatically extracted with the Python's built-in `ast` library. We then convert each API call into its canonical form, where each layer/function has a unique name, and all of its arguments are converted to keyword arguments. Finally, we extract all *API keywords* from the canonicalized API call, where an *API keyword* is the name of a layer/function or the name of a keyword argument.
2. Transform the program into its *code skeleton* by replacing each API keyword occurrence with a distinct placeholder.
3. Transpile the code skeleton, where all API keywords are replaced by placeholders, into the target DL framework using a pretrained big LM (e.g., Codex).
4. Look up each API keyword in the *API keyword dictionary*, and replace each keyword with its translation. To generate the API keyword dictionary, we first learn the API embeddings using domain-adversarial training based on contextual embeddings extracted by PyBERT (a BERT pretrained on Python code and then fine-tuned on deep learning code). Next, we calculate the cosine similarity between the embedding vectors. Then we generate the API keyword dictionary using a hierarchical algorithm.

5. Put each API keyword back into the transpiled code skeleton to generate the final output.

We describe each of these steps next in detail.

## 2.1 CANONICALIZATION & API KEYWORD EXTRACTION

We first parse the source code into an *abstract syntax tree (AST)* with the Python `ast` module. Then, canonicalization and API call extraction are applied to the AST.

**Canonicalization.** We canonicalize each API call using the following steps during both domain-adversarial training (Section 2.3) and inference. Each step involves a recursive AST traversal.

1. Unify the different import aliases of each module into the most commonly used name in the training dataset. For example, `torch.nn` is converted to `nn`.
2. Unify different aliases of each layer/function in a DL library into the name in which it was defined. We detect and resolve each alias by looking at its `__name__` attribute, which stores the callable's original name in its definition.[1] For example, `layers.MaxPool2D` is converted to `layers.MaxPooling2D`.
3. Convert each positional argument of an API call into its equivalent keyword argument. Sort all keyword arguments according to the order defined in the function signature. This is done by linking the arguments of each API call to the parameters of its API signature using the `bind` method from Python's `inspect` module.[2]

**API keyword extraction.** We define *API keyword* as the name of a layer/function or the name of a keyword argument. Once the input code is canonicalized, we locate each API keyword in the AST and then unparse the AST into the canonicalized source code.

## 2.2 SKELETAL CODE TRANSPILATION

After canonicalizing the source program, ADELT then replaces all API keywords with a placeholder, turning the source program into its *code skeleton*. Each placeholder has textual form `PLACEHOLDER_i`, where $i = 1, 2, 3, \ldots$. The code skeleton is then translated by Codex using few-shot prompting. The full prompt for this step is shown in Appendix A.5.

## 2.3 DOMAIN-ADVERSARIAL TRAINING

Once the code skeleton is transpiled, we then transpile API keywords. We train the aligned embeddings of API keywords in a domain-adversarial setting. In Section 2.4, the embeddings will be used to generate a dictionary that maps an API keyword of the source deep learning framework $\mathcal{X}^{(1)}$ to an API keyword in the target DL framework $\mathcal{X}^{(2)}$.

Figure 2 illustrates the domain-adversarial approach of ADELT, and Algorithm 1 shows the pseudocode. A generator maps the contextual representations extracted by PyBERT into hidden states (line 5-8). The alignment of hidden states from different DL frameworks is enforced by the adversarial loss induced by the discriminator (line 17-21), so that output embeddings learned with these hidden states (line 11-14) are also aligned. Next, we describe each step in detail:

Each training example is a pair of API keyword occurrences with their context in the training corpus, denoted by $(x^{(1)}, x^{(2)})$. Each keyword occurrence $x^{(l)}$ is tokenized and encoded as multiple *byte pair encoding (BPE)* (Sennrich et al., 2016) tokens. In our unsupervised setting, $x^{(1)}$ and $x^{(2)}$ are independent samples from $\mathcal{X}^{(1)}$ and $\mathcal{X}^{(2)}$ in the training dataset, respectively, and they are not necessarily translations of each other.

**PyBERT.** *PyBERT* is our pretrained Transformer (Vaswani et al., 2017; Devlin et al., 2019) for Python code (Feng et al., 2020; Kanade et al., 2020; Roziere et al., 2021). Given a sequence of BPE tokens that represent an API keyword with its context $x^{(l)}$, PyBERT outputs a sequence of vectors—one vector in $\mathbb{R}^{d_b}$ for each token, where $d_b$ is the hidden dimension size of PyBERT. We

---

[1] `https://docs.python.org/3/reference/datamodel.html#the-standard-type-hierarchy`
[2] `https://docs.python.org/3/library/inspect.html#inspect.Signature.bind`

---

**Algorithm 1** Pseudo-code for domain-adversarial training.

```
1  for (x_1, y_1), (x_2, y_2) in loader:
2    # N samples from X_1, X_2 respectively
3    # y_1, y_2: API keyword ids
4
5    h_1 = B(x_1).detach()  # contextual embedding
6    h_2 = B(x_2).detach()  # no gradient to PyBERT
7    z_1 = G(h_1)  # generator hidden states
8    z_2 = G(h_2)  # z_1, z_2: N x d
9
10   # dot product of z_l and output embeddings
11   logits_1 = mm(z_1, E_1.view(d, m_1))
12   logits_2 = mm(z_2, E_2.view(d, m_2))
13   L_CE_1 = CrossEntropyLoss(logits_1, y_1)
14   L_CE_2 = CrossEntropyLoss(logits_2, y_2)

15
16   # discriminator predictions
17   pred_1 = D(z_1)
18   pred_2 = D(z_2)
19   labels = cat(zeros(N), ones(N))
20   L_D = CrossEntropyLoss(pred_1, labels)
21   L_G = CrossEntropyLoss(pred_2, 1 - labels)
22
23   # joint update of G and E_l
24   #   to minimize L_CE_l
25   optimize(G + E_1 + E_2, L_CE_1 + L_CE_2)
26   optimize(D, L_D)  # train the discriminator
27   optimize(G, L_G)  # train the generator
```

---

B: PyBERT used as the contextual embedder. G, D: the generator $\mathcal{G}$ and the discriminator $\mathcal{D}$.
E_l: a $d$ by $m_l$ matrix, where the $i$-th column vector is the output embedding of API keyword $w_i^{(l)}$.
mm: matrix multiplication; cat: concatenation

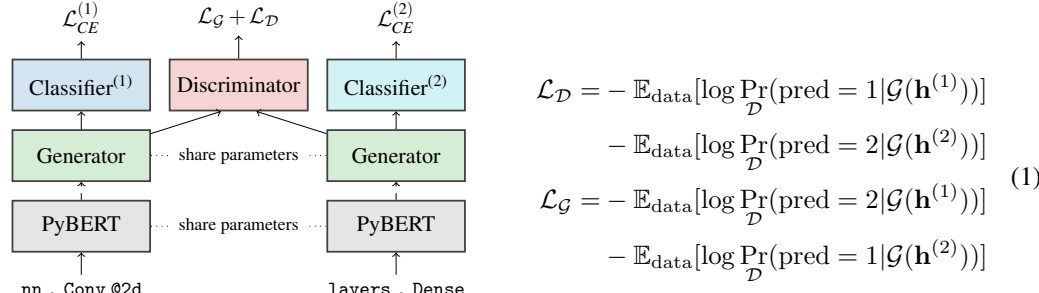

$$\mathcal{L}_{\mathcal{D}} = - \mathbb{E}_{\text{data}}[\log \Pr_{\mathcal{D}}(\text{pred} = 1|\mathcal{G}(\mathbf{h}^{(1)}))]$$
$$- \mathbb{E}_{\text{data}}[\log \Pr_{\mathcal{D}}(\text{pred} = 2|\mathcal{G}(\mathbf{h}^{(2)}))]$$
$$\mathcal{L}_{\mathcal{G}} = - \mathbb{E}_{\text{data}}[\log \Pr_{\mathcal{D}}(\text{pred} = 2|\mathcal{G}(\mathbf{h}^{(1)}))]$$
$$- \mathbb{E}_{\text{data}}[\log \Pr_{\mathcal{D}}(\text{pred} = 1|\mathcal{G}(\mathbf{h}^{(2)}))] \quad (1)$$

Figure 2: ADELT's domain-adversarial training with contextual embeddings from a PyBERT. The generator and the PyBERT are shared between different DL frameworks. We do not fine-tune the PyBERT during adversarial training.

$$\mathcal{L}_{\text{CE}}^{(l)} = -\mathbb{E}_{(x,y)\sim\text{data}^{(l)}}\left[\log \frac{\exp(\mathbf{z} \cdot \mathbf{e}_y^{(l)})}{\sum_{k=1}^{m^{(l)}} \exp(\mathbf{z} \cdot \mathbf{e}_k^{(l)})}\right] \quad (2)$$

average-pool all BPE tokens of the keyword and get a single $d_b$-dimensional vector as the contextual embedding $\text{PyBERT}(x^{(l)})$ of the API keyword. We denote the contextual embedding of $x^{(1)}, x^{(2)}$ by $\mathbf{h}^{(1)}, \mathbf{h}^{(2)}$ respectively.

**Generator and discriminator.** We define two multi-layer perceptrons, a generator and a discriminator. A generator $\mathcal{G}$ encodes the contextual embeddings $\mathbf{h}^{(1)}, \mathbf{h}^{(2)}$ into hidden states $\mathbf{z}^{(1)}, \mathbf{z}^{(2)} \in \mathbb{R}^d$, and a discriminator $\mathcal{D}$ is trained to discriminate between $\mathbf{z}^{(1)}$ and $\mathbf{z}^{(2)}$. The generator is trained to prevent the discriminator from making accurate predictions, by making $\mathcal{G}(\text{PyBERT}(\mathcal{X}^{(1)}))$ and $\mathcal{G}(\text{PyBERT}(\mathcal{X}^{(2)}))$ as similar as possible. Our approach is inspired by domain-adversarial training (Ganin et al., 2016), where domain-agnostic representations of images or documents are learned for domain adaptation. In our case, a domain is represented by a DL framework.

Formally, we define the probability $\Pr_{\mathcal{D}}(\text{pred} = l|\mathbf{z})$ that a hidden state $\mathbf{z}$ is from the DL framework $l$ predicted by the discriminator. Note that $\mathbf{z}^{(1)} = \mathcal{G}(\mathbf{h}^{(1)})$ and $\mathbf{z}^{(2)} = \mathcal{G}(\mathbf{h}^{(2)})$. The discriminator loss and the generator loss are computed as the binary cross entropy against the true label and the reversed label, respectively, as shown in Equation (1).

**Output embeddings.** Our goal is to learn an embedding for each API keyword, but the contextual embedding of each keyword occurrence varies with its context. So we instead train a $d$-dimensional vector $\mathbf{e}_i^{(l)}$ for each API keyword $w_i^{(l)}$, such that $\mathbf{e}_i^{(l)}$ is similar to the generator hidden states $\mathbf{z}_j^{(l)}$ of this keyword's occurrences and dissimilar to the hidden states $\mathbf{z}_k^{(l)}$ of any other keyword's occurrences. $\mathbf{e}_i^{(l)}$ is considered the *output embedding* of the API keyword $w_i^{(l)}$. With similarity computed using

dot product, our optimization objective is shown in Equation (2), equivalent to the cross-entropy loss of $m^{(l)}$-way softmax-based classification.

**Adversarial training.**    During each training iteration, the generator and discriminator are trained successively to minimize $\mathcal{L}_\mathcal{G}$ and $\mathcal{L}_\mathcal{D}$ respectively with mini-batch stochastic gradient descent. Minimizing the adversarial loss equals to minimizing the distance between two distributions of hidden states (Goodfellow et al., 2014). Therefore, the API keywords from the different DL frameworks will be mapped to an aligned embedding space.

Also, we jointly update the generator and the output embeddings to minimize $\mathcal{L}_{\text{CE}}^{(l)}$ with mini-batch SGD. The joint optimization is crucial, as updating the generator to minimize $\mathcal{L}_{\text{CE}}^{(l)}$ ensures that each generator hidden state $\mathbf{z}^{(l)}$ preserves enough information to recover its original API keyword. As a result, the output embeddings $\{\mathbf{e}_i^{(1)}\}_{i=1}^{m^{(1)}}$ and $\{\mathbf{e}_j^{(2)}\}_{j=1}^{m^{(2)}}$ are also aligned, as they are trained with vectors $\mathbf{z}^{(l)}$ from the aligned embedding space.

We do not fine-tune PyBERT during domain-adversarial training, as fine-tuning PyBERT makes the generator disproportionally strong that results in training divergence.

## 2.4    Hierarchical API Dictionary Generation

ADELT calculates a *scoring matrix* using the aligned API keyword embeddings trained in Section 2.3. The entry in the $i$-th row and the $j$-th column of the matrix is the cosine similarity between $w_i^{(1)}$ and $w_j^{(2)}$, denoted by $s_{i,j}$. Given the scoring matrix, we need to generate an API keyword dictionary that maps each API keyword in one deep learning framework to an API keyword in another DL framework.

**Greedy match**    is used to generate a dictionary in word translation of natural languages (Conneau et al., 2018), where each source word is matched to the target word with the highest similarity score.

**Structure of API keywords.**    Unlike words in NL, API keywords are *structured*: API keywords can be classified into two types based on their associated AST node: *callables names* (names of functions or classes), and *parameter names* (names of keyword arguments). In dictionary generation, we do not allow callable names to be translated to parameter names. We only allow parameter names to be translated to callable names when the weight passes a threshold. In this case, this parameter will be dropped and generate a new API call (the last case in Table 2). Another structural property is that the matching of parameters depends on the matching of callables.

**Hierarchical API dictionary generation**    algorithm leverages the structure of API keywords to generate a dictionary: **Step 1.** Consider each callable and its parameters as a group and compute the *group similarity* between each pair of groups, by summing up similarity scores in the greedy matching of parameter names, plus the similarity between two callable names. **Step 2.** Match groups greedily based on group similarity scores calculated in step 1.

## 3    Experiments

We evaluate the effectiveness of ADELT on the task of transpilation between PyTorch, Keras, and MXNet [3] and compare our method with baselines.

## 3.1    Skeletal Code Transpilation

We use Codex (Chen et al., 2021), a GPT model (Brown et al., 2020) finetuned using public GitHub code, to transpile code skeletons. As an autoregressive language model trained on massive web data, Codex can handle translation tasks via prompting with few-shot demonstrations. Our prompt design aligns with Codex's code translation setup, comprising a single input-output example and three instructions to keep placeholders unchanged. Appendix A.5 provides further details on this.

---

[3] We tried to evaluate using JAX (Bradbury et al., 2018). Sadly, JAX is a new DL framework and the GitHub corpus on BigQuery (based on a historical snapshot of GitHub) contains very few (318) examples of JAX.

## 3.2 TRAINING SETUP

**DL corpus.** We consider 4 data sources **GitHub**, **JuiCe** (Agashe et al., 2019), **Kaggle** (Quaranta et al., 2021), and **Web** to build our DL corpus. See Appendix A.1 for details.

We tokenize all Python source code and extract subclasses of `torch.nn.Module`, `keras.layers.Layer`, or `keras.Model`. Then, we canonicalize (section 2.1) the code of each class definition. We byte-pair encode (Sennrich et al., 2016), merge, and deduplicate codes from all sources. Finally, we collect all files into our *DL Corpus* containing 19,796 PyTorch modules, 3,703 Keras layers/models, and 1,783 MXNet modules.

**PyBERT** is our Transformer encoder pretrained with the masked language modeling (MLM) (Devlin et al., 2019) objective on all open-source Python files from the GitHub dataset. We consider two model sizes: PyBERT$_{\text{SMALL}}$ (6-layer, 512-d) and PyBERT$_{\text{BASE}}$ (12-layer, 768-d). Detailed pretraining hyperparameters are described in appendix A.2.

**Adversarial training.** The generator and discriminator of ADELT are multilayer perceptrons. We search the learning rate and batch size according to the unsupervised validation criterion *"average cosine similarity"* (Conneau et al., 2018), which measures the consistency between learned API keyword embeddings and generated keyword translations. Other hyperparameters are set based on previous studies (Conneau et al., 2018) with details described in Appendix A.3.

## 3.3 EVALUATION BENCHMARK

Our method is evaluated through the task of transpiling code snippets from one DL framework to another. We employ heuristics to identify potential match pairs in the corpus and manually curate a robust evaluation benchmark. Detailed methodology and statistics can be found in Appendix A.4.

We report **Exact Match (EM) score** as the main metric. For each code snippet, a model's transpilation is considered to be an exact match if and only if it is exactly equivalent to the ground truth. The EM score is the number of exact matches divided by the number of examples in the eval set. We also report a more forgiving metric, the F1 score, which quantifies the overlap between the predicted and ground truth outputs. In this context, we treat each prediction or ground truth as a bag of function calls. For each test case, we determine the number of exactly matched calls $n_{\text{match}}$, predicted calls $n_{\text{pred}}$, and ground truth calls $n_{\text{truth}}$. We define the F1 score for a particular example as $2n_{\text{match}}/(n_{\text{pred}} + n_{\text{truth}})$, and report the average F1 scores across all test cases.

## 3.4 EVALUATION OF SKELETAL CODE TRANSPILATION

Transpiling code skeletons of DL programs is an easy task, and Codex easily learned transpilation patterns via few-shot prompting. In our evaluation benchmark, the exact match score of skeletal code transpilation using Codex is 100%.

## 3.5 COMPARISON WITH OTHER METHODS

We compare ADELT using PyBERT$_{\text{SMALL}}$ and ADELT using PyBERT$_{\text{BASE}}$ with the following baselines. We run all methods 5 times with random seeds `[10, 20, 30, 40, 50]`, and report the arithmetic average of all metrics.

**End-to-end language models.** We compare ADELT with end-to-end few-shot LLM baselines, including GPT-3, Codex, and GPT-4, where the entire piece of source code, instead of the code skeleton, is fed into the LM to generate the transpiled target program. For source-to-source translation, we randomly give the LLM 5 examples as demonstrations. The prompt design is similar to the code translation setup of Codex. Details are shown in Appendix A.6.

**Edit distance.** We consider a rule-based baseline where we use edit distance (Levenshtein, 1966) as the similarity measure between API keywords, in place of the similarity measures calculated from learned embeddings. We apply hierarchical API dictionary generation exactly as what we do in ADELT. We report the result of both cased and uncased setups for edit distance calculation.

Table 1: **Comparison between ADELT and other methods** on source-to-source transpilation. "ADELT (Small)" is ADELT with PyBERT$_{\text{SMALL}}$ and "ADELT (Base)" is ADELT with PyBERT$_{\text{BASE}}$. There are two numbers in each table cell: the first one is for transpiling PyTorch to the other framework (Keras or MXNet), and the second one is for transpiling the other framework to PyTorch. Each number is the average of 5 runs with different random seeds.

| | PyTorch-Keras | | PyTorch-MXNet | |
|---|---|---|---|---|
| | F1 | EM | F1 | EM |
| GPT-3 (Brown et al., 2020) | 26.6 32.1 | 22.5 26.0 | 25.8 32.8 | 23.4 25.1 |
| Codex (Chen et al., 2021) | 59.9 67.1 | 51.5 54.6 | 57.4 69.0 | 53.2 56.5 |
| GPT-4 | 67.7 74.9 | 55.6 64.7 | 60.3 71.8 | 54.0 60.2 |
| Edit Distance (Cased) | 31.2 30.1 | 20.3 16.8 | 37.7 35.7 | 22.9 21.1 |
| Edit Distance (Uncased) | 23.9 30.1 | 12.5 16.8 | 30.8 36.0 | 18.5 20.1 |
| ADELT (Small) | 79.0 76.7 | 70.7 67.5 | 76.7 70.6 | 66.5 62.9 |
| ADELT (Base) | **83.4 79.3** | **73.0 71.5** | **80.0 72.1** | **70.0 63.7** |

Table 2: **Examples from the evaluation dataset of the PyTorch-Keras transpilation task and the Keras-PyTorch transpilation task.** We show the source code, ground truth target code, and the outputs from Codex, ADELT, and ADELT +. ✓: the output is the same or equivalent to the ground truth. ✓: the output contains an equivalent of the ground truth, but it also contains incorrect extra code. ✗: the output is incorrect.

| Source | `nn.Conv2d(64, 128, 3)` | Source | `nn.Embedding(vocab_size, embed_dim)` |
|---|---|---|---|
| Truth | `layers.Conv2D(filters=128,`
`    kernel_size=3)` | Truth | `layers.Embedding(input_dim=vocab_size,`
`    output_dim=embed_dim)` |
| Codex ✓ | `layers.Conv2D(128, 3)` | Codex ✓ | `layers.Embedding(vocab_size, embed_dim)`
`self.position_emb = layers.Embedding(...)` |
| ADELT ✓ | `layers.Conv2D(filters=128,`
`    kernel_size=3)` | ADELT ✗ | `layers.Embedding(`
`    embeddings_initializer=embed_dim)` |
| Source | `nn.MultiheadAttention(`
`    model_dim, num_heads=num_heads,`
`    dropout=attn_dropout)` | Source | `in_dim = 256`
`out_dim = 512`
`layers.Dense(out_dim, activation='relu')` |
| Truth | `layers.MultiHeadAttention(`
`    num_heads=num_heads,`
`    key_dim=model_dim,`
`    dropout=attn_dropout)` | Truth | `in_dim = 256`
`out_dim = 512`
`nn.Linear(in_dim, out_dim)`
`nn.ReLU()` |
| Codex ✗ | `layers.MultiHeadAttention(`
`    model_dim, num_heads,`
`    dropout=attn_dropout)` | Codex ✗ | `in_dim = 256`
`out_dim = 512`
`nn.Linear(in_dim, out_dim)` |
| ADELT ✓ | `layers.MultiHeadAttention(`
`    num_heads=num_heads,`
`    key_dim=model_dim,`
`    dropout=attn_dropout)` | ADELT ✗ | `in_dim = 256`
`out_dim = 512`
`nn.Linear(in_features=in_dim,`
`    out_features=out_dim)` |
| | | ADELT + ✓ | `in_dim = 256`
`out_dim = 512`
`nn.Linear(in_features=in_dim,`
`    out_features=out_dim)`
`nn.ReLU()` |

The result is shown in Table 1. **ADELT consistently outperforms other methods with respect to all metrics**, and it benefits from a larger pretrained PyBERT embedder. Moreover, even if LLMs used more examples for few-shot supervision, ADELT still consistently outperforms the end-to-end GPT-4 baseline.

### 3.6 CASE STUDIES

Table 2 shows four examples of PyTorch-Keras transpilation together with hypotheses of Codex and ADELT (Base). Both Codex and ADELT transpile the `nn.Conv2d` to Keras correctly by dropping the first argument `in_channels`. ADELT does not translate the parameter names of `nn.Embedding` to `input_dim` and `output_dim` correctly, while Codex does. However, we notice that Codex sometimes

relies on the argument ordering heuristic. In the example of `nn.MultiheadAttention`, where parameters have a different ordering in Keras than in PyTorch, Codex generates the wrong translation, but ADELT successfully constructs the correct mapping between parameters.

Also, in the `nn.Embedding` example, Codex continues to generate code about "positional embeddings" after finishing transpilation. The extra code generated by Codex is relevant to the context.[4] Still, the extra code should not be part of the translation. We have tried various ways to make Codex follow our instructions (see Appendix A.6 for details). However, because Codex is an end-to-end neural language model, our means of changing its predictions are limited, and the result is highly indeterministic. In the end, Codex still occasionally generates extra arguments or unneeded statements.

On the other hand, we decouple neural network training from the transpilation algorithm. ADELT transpiles between deep learning frameworks using deterministic keyword substitution based on a learned API keyword dictionary. The transpiled code is always syntactically correct. If a mistake is found in the dictionary (e.g., the `nn.Embedding` example in Table 2), it can be corrected by simply modifying the dictionary.

Correcting the API keyword dictionary by humans requires much less effort than building the dictionary manually from scratch, as ADELT generates a high-quality dictionary. Developers can even add additional rules to the transpiler. The flexibility of our decoupled design makes ADELT far easier to be integrated into real-world products than end-to-end neural translators/LMs are.

The last case in Table 2 shows an example where an API call (`layers.Dense` with `activation="relu"`) should be transpiled to two calls (`nn.Linear` and `nn.ReLU`). One-to-many mapping is rare in transpilation between deep learning frameworks, but the capability to model such mapping reflects the generality of a transpiler to other APIs. Both ADELT and Codex fail to solve this example because this usage is rarely seen in the training data. Still, if we train ADELT on an additional synthetic dataset ("ADELT +" in Table 2. See Appendix A.9 for details), it successfully solves this case, showing that our method can model one-to-many mappings when enough training data is available.

## 3.7 ABLATION STUDIES

We conduct ablation studies on PyTorch-Keras transpilation to validate the contribution of each part of ADELT. We consider both source-to-source transpilation and API keyword translation. **API keyword translation** involves retrieving the translation of given API keywords. We create a high-quality dictionary by manually translating the first 50 most frequent API keywords in PyTorch and Keras, respectively. Following the standard practice of word translation, we measure how many times the correct translation of a source word is retrieved (**precision@$k$** for $k = 1, 5$) and the **mean reciprocal rank** of the correct translation (MRR). The results are shown in Table 3.

**Necessity of contextual embeddings.** In *"w/o PyBERT"*, we replace PyBERT with Word2Vec (Mikolov et al., 2013) embeddings of the same dimensions $d_b$ trained on the same corpora. The result in Table 3 shows that this change significantly harms the performance of ADELT. This justifies the use of PyBERT, a high-quality pretrained representation of API keywords that can capture their contexts.

**Contribution of adversarial loss.** In *"w/o Adv Loss"*, we remove the adversarial loss during training. Instead, we only train the generator and the output embeddings with the cross-entropy loss in Equation (2). The result in Table 3 shows that adversarial training contributes ∼6 pts in source-to-source transpilation, showing the effectiveness of adversarial training.

**Comparison of similarity measures.** By default, ADELT uses cosine similarity as the similarity measure for API dictionary generation. Table 3 shows the results of using dot product (inner). Measures based on cosine similarity outperforms dot product by a small margin. This fact implies that the performance of ADELT is insensitive to the choice of similarity measure.

---

[4]The definition of positional embeddings usually follows the definition of word embeddings (`nn.Embedding(vocab_size, ...)`) in the source code of a Transformer model.

Table 3: **Ablation study results.** By default, ADELT is trained with the adversarial loss on contextual embeddings extracted by PyBERT, and then a dictionary is generated based on cosine similarity scores. We change one component of ADELT (Small) or ADELT (Base) in each experiment to assess its contribution.

| | Keyword | | | | | Source Code | |
| --- | --- | --- | --- | --- | --- | --- | --- |
| | P@1 | | P@5 | | MRR | | F1 |
| ADELT (Small) | 82.9 | 90.0 | **91.7** | **97.7** | 87.0 | 94.0 | 79.0 | 76.7 |
| ADELT (Base) | **87.1** | 90.0 | **91.7** | **97.7** | **89.7** | 94.0 | **83.4** | **79.3** |
| *Domain-adversarial training* | | | | | | | | |
| w/o PyBERT (Small) | 52.1 | 63.6 | 70.0 | 85.9 | 60.5 | 72.8 | 37.2 | 43.0 |
| w/o PyBERT (Base) | 45.0 | 54.6 | 70.4 | 80.0 | 56.8 | 66.0 | 33.0 | 36.3 |
| w/o Adv Loss (Small) | 80.4 | 88.6 | 90.0 | **97.7** | 85.3 | 93.1 | 65.8 | 73.6 |
| w/o Adv Loss (Base) | 86.3 | 90.5 | **91.7** | **97.7** | 89.3 | 94.3 | 78.2 | 72.3 |
| *Measure for dictionary generation* | | | | | | | | |
| Inner Product (Small) | 81.3 | 79.6 | **91.7** | 90.0 | 86.3 | 85.4 | 74.6 | 73.2 |
| Inner Product (Base) | 85.4 | **93.2** | **91.7** | **97.7** | 88.8 | **95.7** | 80.2 | 78.8 |

## 4 RELATED WORK

**Source-to-source transpilation.** Classical source-to-source transpilers use supervised learning. Nguyen et al. (2013) and Karaivanov et al. (2014) develop Java-C# transpilers using parallel corpora of open-source code. The dependency on parallel corpora renders these methods inapplicable to transpilation between deep learning frameworks, as parallel corpora are difficult to get.

Drawing inspiration from unsupervised neural machine translation (NMT) (Artetxe et al., 2018), recent advancements have made unsupervised programming language translation possible (Lachaux et al., 2020). Such approaches, however, require vast amounts of in-domain unlabeled corpora, as evidenced by Lachaux et al. (2020) and Roziere et al. (2022), who utilized 744GB of GitHub source code and a dataset of 333k curated Java functions respectively. The scarcity of online deep learning code hinders their effectiveness for transpilation between DL frameworks, as we illustrate in Section 3.5.

**Language models are few shot learners.** GPT-3 (Brown et al., 2020) is a language model with 175B parameters trained on massive web crawl data. GPT-3 can be applied to many NLP tasks without any gradient updates or fine-tuning, with tasks and few-shot demonstrations specified purely via text interaction with the model. Codex (Chen et al., 2021) is a GPT-3 fine-tuned on publicly available code from GitHub, specialized for code generation tasks. GPT-4 is a LLM proficient in both code and NL trained using instruction finetuning. In constrast, the code generation step of ADELT is keyword substitution instead of autoregressive generation. ADELT outperforms GPT-3, Codex, and GPT-4 in PyTorch-Keras transpilation.

**Adversarial learning & cross-lingual word embedding.** Conneau et al. (2018) uses domain-adversarial (Ganin et al., 2016) approach to align the distribution of two word embeddings, enabling natural language word translation without parallel data. The domain-adversarial training in ADELT is inspired by their approach, but we align the distributions of the hidden states of *keyword occurrences* instead of API keyword embeddings.

## 5 CONCLUSION

We presented ADELT, a code transpilation algorithm for deep learning frameworks. ADELT formulates the transpilation problem as API keyword mapping, and uses domain-adversarial training to generate the map. Using our collected Pytorch-Keras and PyTorch-MXNet benchmarks, our evaluation shows that ADELT can significantly outperform state-of-the-art transpilers.

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

# A APPENDIX

## A.1 DATA SOURCE FOR OUR CORPUS

We consider 4 data sources **GitHub**, **JuiCe** (Agashe et al., 2019), **Kaggle** (Quaranta et al., 2021), and **Web** to build our DL corpus:

- **GitHub**: The GitHub public dataset available on Google BigQuery.[5] We keep py and ipynb files that contain torch, keras, or mxnet in the main and master branch of the repository. (2.5GB after filtering)

- **JuiCe**: A code generation dataset (Agashe et al., 2019) based on ipynb files from GitHub. JuiCe contains many files absent in the public dataset on Google BigQuery, since the latter is a selected subset of GitHub (10.7GB of clean Python code)

- **Kaggle**: All files in KGTorrent (Quaranta et al., 2021), a dataset of Jupyter Notebooks from Kaggle.[6] (22.1GB of clean Python code)

- **Web**: Python code snippets in web pages of relevant websites. We extract texts in <pre> tags of HTML files scraped from Stack Overflow[7] (60MB) and PyTorch Forums[8] (25MB).

## A.2 PYBERT PRE-TRAINING HYPERPARAMETERS AND IMPLEMENTATION DETAILS

The models are pre-trained with the RoBERTa (Liu et al., 2019) pipeline in fairseq[9] codebase. We pre-train each PyBERT on the GitHub dataset. On a NVIDIA DGX-2, it takes 8.2 hours and 23.1 hours to train PyBERT$_{\text{SMALL}}$ and PyBERT$_{\text{BASE}}$, respectively.

table 4 shows the pre-training hyperparemters of PyBERT$_{\text{SMALL}}$ and PyBERT$_{\text{BASE}}$. We first pre-train each model on the Github dataset and then fine-tune it on our canonicalized PyTorch-Keras corpus. The learning rate is decayed according to the inverse square root schedule. We do not use early stopping — we use the last PyBERT checkpoint in ADELT.

Table 4: Pre-training hyperparameters of PyBERT

| Hyperparameter | PyBERT$_{\text{SMALL}}$ | PyBERT$_{\text{BASE}}$ |
|---|---|---|
| Number of layers | 6 | 12 |
| Hidden size $d_b$ | 512 | 768 |
| FFN inner hidden size | 2048 | 3072 |
| Attention heads | 8 | 12 |
| Attention head size | 64 | 64 |
| Dropout | 0.1 | 0.1 |
| Attention dropout | 0.0 | 0.0 |
| FFN dropout | 0.0 | 0.0 |
| Adam $\beta_1$ | 0.9 | 0.9 |
| Adam $\beta_2$ | 0.98 | 0.98 |
| Adam $\epsilon$ | 1e-6 | 1e-6 |
| Weight decay | 0.01 | 0.01 |
| Gradient clipping | - | - |
| Peak learning rate | 5e-4 | 5e-4 |
| Batch size | 2,048 | 2,048 |
| Warmup steps | 10,000 | 10,000 |
| Total steps | 125,000 | 125,000 |

---

[5]https://console.cloud.google.com/marketplace/details/github/github-repos

[6]https://kaggle.com

[7]https://stackoverflow.com/

[8]https://discuss.pytorch.org/

[9]https://github.com/facebookresearch/fairseq

### A.3 DOMAIN-ADVERSARIAL TRAINING HYPERPARAMETERS

The generator and the discriminator of ADELT are multilayer perceptrons. The activation function is ReLU for the generator and Leaky-ReLU for the discriminator. Dropout and label smoothing are applied for regularization. We train our generator, discriminator, and API keyword embeddings with Adam (Kingma and Ba, 2017) on 1,536,000 samples. There is a linear learning rate warmup over the first 10% of steps, and then we set the LR according to the invert square root decay rule. The learning rate scheduler uses linear warmup and inverse sqrt decay. The maximum learning rate is searched from [2e-4, 5e-4, 1e-3], and the batch size is searched from [64, 128, 256] The peak learning rate and the batch size are searched according to the unsupervised validation criterion *"average cosine similarity"* (Conneau et al., 2018) of the generated dictionary, which quantifies the consistency between the learned API keyword embeddings and the generated keyword translations. We set other hyperparameters according to prior works (Conneau et al., 2018), shown in table 5 (top). The learning rates and the batch sizes selected in the hyperparameter search are shown in table 5 (bottom). The total number of training steps is "total samples" (1,536,000) divided by the searched batch size, which is 6,000 steps for ADELT (Small) and 12,000 steps for ADELT (Base).

Table 5: The hyperparameters of domain-adversarial training

| | |
|---|---|
| Generator activation | ReLU |
| Generator hidden size | 2,048 |
| Generator layers | 1 |
| Discriminator hidden size | 2,048 |
| Discriminator layers | 1 |
| Discriminator activation | LeakyReLU |
| Discriminator LeakyReLU slope | 0.2 |
| Dropout | 0.1 |
| Label smoothing | 0.2 |
| Warmup step ratio | 10% |
| Adam $\beta_1$ | 0.9 |
| Adam $\beta_2$ | 0.999 |
| Adam $\epsilon$ | 1e-8 |
| Weight decay | 0.001 |
| Discriminator iterations per step | 1 |
| Total samples | 1,536,000 |
| Peak learning rate (Small) | 2e-4 |
| Batch size (Small) | 128 |
| Peak learning rate (Base) | 5e-4 |
| Batch size (Base) | 256 |

### A.4 EVALUATION DATA COLLECTION

Our evaluation uses a parallel corpus of 100 examples derived from two distinct methods:

**Heuristically Identified Parallel Examples:**   We sourced open-source projects on GitHub that benchmark various deep learning frameworks by replicating identical neural network architectures across those frameworks. Our parallel pair identification process relied on a heuristic comparing Python class names. Criteria for selection included pairs of PyTorch modules and Keras models/layers that (a) possessed identical class names and (b) achieved a BLEU score above 65. Following heuristic selection, we refined these pairs through manual extraction of code segments containing deep learning API calls, resulting in a corpus of 50 parallel examples. Then, human experts manually transpile those 50 PyTorch examples to MXNet, resulting in a corpus of 50 parallel examples for evaluating PyTorch-MXNet transpilation.

**Expert-Transpiled Examples:**   The second set of 50 examples was assembled by selecting PyTorch module definitions from GitHub repositories with more than 1,000 stars and asking human experts to convert them into the Keras framework. The resulting PyTorch-Keras pairs tend to be longer and more challenging.

## A.5 DETAILS OF SKELETAL CODE TRANSPILATION

Table 6 shows by example how we transpile skeletal codes using Codex few-shot prompting.

1. Each API keyword in the canonicalized source program is replaced with an distinct place-holder, numbered from 1 to $n$ (the number of API keywords). The program after this step is called the code skeleton of the source program.

2. We append the code skeleton to the natural language prompt, *# Translate from PyTorch to Keras*, and four input-output pairs. The first three input-output pairs prompt the model to keep placeholders unchanged during transpilation. Our experiments show that three input-output pairs are required for 100% skeletal code transpilation correctness. Also, Codex can generalize to an arbitrary number of placeholders even if only three is given. The last input-output pair is a real example of PyTorch-Keras skeletal code transpilation.

3. This entire piece of input is fed into Codex, and Codex will complete this input by generating tokens after `# Keras`. The output of Codex is considered as the code skeleton of the target program.

4. Each placeholder is replaced with the API keyword in the target DL framework, by querying each API keyword before replacement (step 1) in the API keyword dictionary learned with ADELT.

If the number of placeholders in the source skeleton and the number of placeholders in Codex's output do not match, it is considered a failed example in evaluation. However, in practice, the success rate of skeletal code transpilation is 100% in our experiments. We attribute that to the fact that skeletal code in DL programs, in comparison to arbitrary Python code, tend to be high structured with fairly predictable import statements, constructors, and how the different DL layers are constructed and connected to each other.

## A.6 EVALUATION SETUP OF LLMS

Following the practices in Brown et al. (2020) and Chen et al. (2021), we use the *"completion"* endpoint of GPT-3 or Codex and the *"chat"* endpoint of GPT-4 for transpilation. We input some text as a prompt with few-shot demonstrations, and the model will generate a completion that attempts to match the prompt. table 7 shows two examples illustrating how we leverage GPT-3 or Codex for our task. table 8 shows two examples illustrating how we leverage GPT-4 for our task.

For source-to-source transpilation, prompt engineering is straightforward. In the PyTorch-Keras transpilation example, we tell the model to "`# Translate from PyTorch to Keras`" and then give 5 demonstrations from our evaluation dataset. Next, we input a piece of source code and "`# Keras`" and let the model generate a code completion starting from the following line. For chat models like GPT-4, we formulate the prompt in Markdown format, and extract the contents first Markdown code block as the model's output. To prevent answers from being leaked to the language model, we do not allow any demonstration to share common API functions with the current evaluation example.

Prompt engineering of API keyword translation is trickier because there are two types of keywords. We represent callable names by one line containing its textual representation, and we represent parameter names by two lines, where the first line is the name of the callable that the parameter belongs to, and the second line is the name of the parameter. We give 10 demonstrations from our evaluation dataset.

Although GPT-3 and Codex have strong capabilities in generating code related to our prompt, we find that they sometimes fail to follow our instructions to transpile between deep learning frameworks. We discuss this problem in section 3.5. We try several approaches to mitigate this issue:

1. Use the *Instruct* version[10] of GPT-3/Codex: `text-davinci-002` and `code-davinci-002`.

2. Add a prefix to the input prompt based on simple rules. For example, if the source code starts with `nn.` in PyTorch, add `layers.` to the prompt and let the model generate a code completion after it. This trick is applicable to two examples shown in table 7.

---

[10]`https://help.openai.com/en/articles/5832130-what-s-changed-with-engine-names-and-best-practices`

3. Mask the logits of tokens that usually leads to irrelevant generations. Specifically, we find that the model tends to generate irrelevant extra code after a line break or random comments. So we add a bias of -100 to the logits of the hash mark "#". We also add a bias of -100 to the logits of the line break if the source code contains no line breaks.

We find that these measures significantly improve the performance of GPT-3 and Codex on deep learning transpilation. All results of GPT-3 and Codex reported in section 3.5 are from the LMs with all these tricks turned on.

Conversely, GPT-4 did not exhibit similar issues, and given that it does not support logits masking, we engaged it directly using the prompts in Table 8 on `gpt-4-0314` without additional alterations.

## A.7 CROSS-DOMAIN LOCAL SCALING (CSLS)

Cross-Domain Local Scaling (CSLS) is a similarity measure for creating a dictionary based on high-dimensional embeddings. CSLS was proposed by Conneau et al. (2018) for word translation between natural languages. Empirical results by Conneau et al. (2018) show that using a pairwise scoring matrix (e.g. cosine similarity, dot product) in dictionary generation suffers from the *hubness* problem (Radovanović et al., 2010), which is detrimental to generating reliable matching pairs as some vectors, dubbed *hubs*, are the nearest neighbors to many other vectors according to $s$, while others (anti-hubs) are not nearest neighbors of any point. This problem is observed in various areas (Jegou et al., 2010; Dinu et al., 2015). CSLS is proposed to mitigate the hubness problem.

We also conduct an experiment to verify the effectiveness of CSLS in API keyword translation between deep learning frameworks. Specifically, we denote by $\mathcal{N}_s^{(l)}(w)$ the *neighborhood* of API keyword $w$, a set consisting of $K$ elements with the highest similarity scores with $w$ in DL framework $\mathcal{X}^{(l)}$. We calculate the average similarity score of $w_i^{(1)}$ to its neighborhood in DL framework $\mathcal{X}^{(2)}$ and denote it by $r_i^{(2)}$. Likewise, we denote by $r_j^{(1)}$ the average similarity score of $w_j^{(2)}$ to its neighborhood in DL framework $\mathcal{X}^{(1)}$. Then we define a new similarity measure CSLS of $w_i^{(1)}$ and $w_i^{(2)}$ by subtracting $r_i^{(2)}$ and $r_j^{(1)}$ from their (doubled) similarity score $s_{i,j}$, as shown in eq. (3).

$$
\begin{aligned}
r_i^{(2)} &= \frac{1}{K} \sum_{k \in \mathcal{N}_s^{(2)}(w_i^{(1)})} s_{i,k} \\
r_j^{(1)} &= \frac{1}{K} \sum_{k \in \mathcal{N}_s^{(1)}(w_j^{(2)})} s_{k,j} \\
\text{CSLS}_{i,j} &= 2s_{i,j} - r_i^{(2)} - r_j^{(1)}
\end{aligned}
\tag{3}
$$

CSLS can be induced from a parameter $K$ and any similarity measure, including dot product and cosine similarity. Intuitively, compared with the score matrix of similarity measure $s$, the score matrix of CSLS assigns higher scores associated with isolated keyword pairs and lower scores of keywords lying in dense areas.

Given the (cosine similarity) scoring matrix scaled by CSLS, we then apply the hierarchical dictionary generation algorithm (section 2.4) to generate the API keyword dictionary. We search $K$ in $\{5, 10, 20\}$ according to the unsupervised evaluation metric, and the result is similar, where $K = 5$ gives a slightly better result. table 9 shows the result of cosine-CSLS compared with cosine similarity.

table 9 shows that replacing cosine similarity with cosine-CSLS-5 does not impact the F1 score of transpiling PyTorch to Keras significantly, but it hurts the F1 score of transpiling Keras to PyTorch. The reason is that the vocabulary of API keywords is smaller than a natural language vocabulary. Hubness is not a problem for generating API keyword dictionaries; instead, penalizing the top-K may hurt the performance when there are relatively few valid candidates (e.g. Keras-to-PyTorch transpilation). Therefore, we do not use CSLS for ADELT.

## A.8 Full Results with Error Bars

table 10 shows full results with error bars for PyTorch-Keras API keyword translation and source-to-source transpilation. The table includes the results of both the main comparison with GPT-3/Codex and ablation studies. We also add the results of GPT-3 and Codex on API keyword translation, where we randomly give the GPT-3 and Codex 10 examples as demonstrations. Details about prompt designs and hyperparameter setup are shown in appendix A.6. We do not calculate precision@5 and mean reciprocal rank for GPT-3 and Codex because the API provided by OpenAI does not support ranking a large number of generations cost-efficiently.

## A.9 ADELT +

We created a new model, ADELT +, which is based on ADELT but trained on a synthetic dataset. Our goal is to evaluate whether our method can generalize to one-to-many mappings of APIs given enough data.

As we discussed in section 2.4, we allow parameter names to be translated to callable names when the weight passes a threshold $\tau$. In this case, this parameter will be dropped and a new API call will be generated. This mechanism allows ADELT to transpile `layers.Dense(...,  activation="relu")` into two layers: `nn.Linear(...)` and `nn.ReLU()`, and similarly `layers.Conv2D(..., activation="relu")` into `nn.Conv2D(...)` and `nn.ReLU()`. However, such cases are rare in transpiling between deep learning frameworks, making it difficult to evaluate our model's ability to transpile one-to-many mappings in practice. Therefore, we create a synthetic dataset, where we replace all consecutive calls of `layers.Dense` and `layers.ReLU` in our dataset with `layers.Dense(..., activation="relu")`, and we replace all consecutive calls of `layers.Conv2D` and `layers.ReLU` with `layers.Conv2D(..., activation="relu")`. Then we train a new model, ADELT +, using our synthetic dataset.

We then evaluated ADELT + using our evaluation dataset. The value of the threshold $\tau$ is set heuristically to 5 (95% of values in the score matrix lies in -7 to 7). table 2 in section 3.6 and table 11 in the appendix show that ADELT + can model one-to-many mappings of APIs. For instance, table 11 shows that ADELT can transpile `layers.Conv2D` with `activation='relu'` into two API calls: `nn.Conv2d` and `nn.ReLU`.

## A.10 More Case Studies

In this section, we select two PyTorch-Keras cases in our evaluation dataset for illustration. They are examples of the average length of all evaluation examples in the evaluation set.

### A.10.1 Case 1

```python
# Source Program
import torch.nn as nn
class BasicBlock(nn.Module):
    def __init__(self, dim):
        super.__init__()
        self.bn1 = nn.BatchNorm2d(dim)
        self.act1 = nn.LeakyReLU(0.2)
        self.conv1 = nn.Conv2d(dim, dim, 3)
        self.pool1 = nn.MaxPool2d(3, 2)

# Transpiled by ADELT
import tensorflow.keras.layers as layers
class BasicBlock(layers.Layer):
    def __init__(self, dim):
        super.__init__()
        self.bn1 = layers.BatchNormalization()
        self.act1 = layers.LeakyReLU(alpha=0.2)
        self.conv1 = layers.Conv2D(filters=dim, kernel_size=3)
        self.pool1 = layers.MaxPooling2D(pool_size=3, stride=2)
```

```
# Ground Truth
import tensorflow.keras.layers as layers
class BasicBlock(layers.Layer):
    def __init__(self, dim):
        super.__init__()
        self.bn1 = layers.BatchNormalization()
        self.act1 = layers.LeakyReLU(0.2)
        self.conv1 = layers.Conv2D(dim, 3)
        self.pool1 = layers.MaxPooling2D(3, 2)
```

### A.10.2 CASE 2

```
# Source Program
import torch.nn as nn
class AttentionBlock(nn.Module):
    def __init__(self, args):
        super().__init__()
        self.attn = nn.MultiheadAttention(
            args.d_model, args.n_heads, dropout=args.att_dropout)
        self.drop1 = nn.Dropout(args.dropout)
        self.norm1 = nn.LayerNorm(args.d_model)
```

```
# Transpiled by ADELT
import tensorflow.keras.layers as layers
class AttentionBlock(layers.Layer):
    def __init__(self, args):
        super().__init__()
        self.attn = layers.MultiHeadAttention(
            num_heads=args.n_heads, key_dim=args.d_model, dropout=args.att_dropout)
        self.drop1 = layers.Dropout(rate=args.dropout)
        self.norm1 = layers.LayerNormalization()
```

```
# Ground Truth
import tensorflow.keras.layers as layers
class AttentionBlock(layers.Layer):
    def __init__(self, args):
        super().__init__()
        self.attn = layers.MultiHeadAttention(
            args.n_heads, args.d_model, dropout=args.att_dropout)
        self.drop1 = layers.Dropout(args.dropout)
        self.norm1 = layers.LayerNormalization()
```

In each case, ADELT makes the correct transpilation. The only textual difference is that ADELT's transpilation only contains keyword arguments while the ground truth still contains positional arguments. However, because the prediction and the ground truth are the same after canonicalization, we consider each case as an exact match during evaluation.

### A.11 DEEP LEARNING TRANSPILATION ACROSS DIFFERENT PROGRAMMING LANGUAGES

In the main paper, all experiments are conducted on Python due to the scarcity of deep learning programs written in other programming languages such as Java or C. Despite that, in this section we show that ADELT is not limited to the same source and target languages by transpiling code written against the PyTorch library in Python 2 to Keras in Python 3.

To do so, we first canonicalize all PyTorch programs into Python 2 and all Keras programs into Python 3. Then we run ADELT on this modified training data to learn the API keyword dictionary. During inference, we transpile the code skeleton with Codex using the prompt shown in Table 12. Besides adding hint words such as "Python2" and "Python3" into the natural language prompt, we also find it necessary to add to the prompt some examples showing differences between Python 2 and Python 3, such as different `print` statements and different integer division operators. As is shown in

Table 12, the skeletal codes were successfully transpiled from Python 2 and Python 3 along with the API keywords.

Table 6: Example inputs we give to Codex for skeletal code transpilation. We also show the expected outputs of the language model.

---

*Canonicalized Source Program*

```python
import torch.nn as nn
dense = nn.Linear(in_features=dim_in, out_features=dim_out, bias=False)
```

---

*Code Skeleton*

```python
import torch.nn as nn
dense = PLACEHOLDER_1(PLACEHOLDER_2=dim_in, PLACEHOLDER_3=dim_out, PLACEHOLDER_4=False)
```

---

*Codex Input*

```python
# Translate from PyTorch to Keras

# PyTorch
PLACEHOLDER_1

# Keras
PLACEHOLDER_1

# PyTorch
PLACEHOLDER_2

# Keras
PLACEHOLDER_2

# PyTorch
PLACEHOLDER_3

# Keras
PLACEHOLDER_3

# PyTorch
import torch.nn as nn
class Model(nn.Module):
    def __init__(self):
        super().__init__()
        self.layer1 = PLACEHOLDER_1(PLACEHOLDER_2=16, PLACEHOLDER_3=32, PLACEHOLDER_4=3)
        self.layer2 = PLACEHOLDER_5()

    def forward(self, x):
        x = self.layer1(PLACEHOLDER_6=x)
        x = self.layer2(PLACEHOLDER_7=x)
        return x

# Keras
import tensorflow.keras.layers as layers
class Model(layers.Layer):
    def __init__(self):
        super().__init__()
        self.layer1 = PLACEHOLDER_1(PLACEHOLDER_2=16, PLACEHOLDER_3=32, PLACEHOLDER_4=3)
        self.layer2 = PLACEHOLDER_5()

    def call(self, x):
        x = self.layer1(PLACEHOLDER_6=x)
        x = self.layer2(PLACEHOLDER_7=x)
        return x

# PyTorch
import torch.nn as nn
dense = PLACEHOLDER_1(PLACEHOLDER_2=dim_in, PLACEHOLDER_3=dim_out, PLACEHOLDER_4=False)

# Keras
```

---

*Expected Codex Output*

```python
import tensorflow.keras.layers as layers
dense = PLACEHOLDER_1(PLACEHOLDER_2=dim_in, PLACEHOLDER_3=dim_out, PLACEHOLDER_4=False)
```

---

*Target Program*

```python
import tensorflow.keras.layers as layers
dense = layers.Dense(units=dim_out, use_bias=False)
```

---

Table 7: Example inputs we give to GPT-3 or Codex for source-to-source transpilation and API keyword translation. We also show the expected outputs of the language models.

| Source-to-Source Transpilation | Keyword Translation |
|---|---|
| ```python
# Translate PyTorch to Keras

# PyTorch
max_len = 512
self.embed_tokens = nn.Embedding(
  n_words, dim_emb)
# Keras
max_len = 512
self.embed_tokens = layers.Embedding(
  n_words, dim_emb, input_length=max_len)

# PyTorch
nn.Linear(dim_in, dim_out)
# Keras
layers.Dense(dim_out)

(2 demonstrations omitted)

# PyTorch
F.log_softmax(logits, dim=-1)
# Keras
tf.nn.log_softmax(logits, axis=-1)

# PyTorch
nn.Conv2d(64, 128, 3)
# Keras
layers.
``` | ```python
# Translate PyTorch to Keras

# PyTorch
F.log_softmax
# Keras
tf.nn.log_softmax

# PyTorch
nn.MaxPool2d
stride
# Keras
layers.MaxPooling2D
strides

(7 demonstrations omitted)

# PyTorch
F.relu
# Keras
tf.nn.relu

# PyTorch
nn.Conv2d
out_channels
# Keras
layers.
``` |
| ```
Conv2D(128, 3)
``` | ```
Conv2D
filters
``` |

Table 8: Example inputs we give to GPT-4 for source-to-source transpilation and API keyword translation. We also show the expected outputs of the language models. Because GPT-4 outputs Markdown texts including both NL and code, we extract contents of the first Markdown code block as the output of the model.

| Source-to-Source Transpilation | Keyword Translation |
|---|---|

```
You are an expert in deep learning.
Transpile PyTorch to Keras:

PyTorch:

```python
max_len = 512
self.embed_tokens = nn.Embedding(
  n_words, dim_emb)
```

Keras:

```python
max_len = 512
self.embed_tokens = layers.Embedding(
  n_words, dim_emb, input_length=max_len)
```

PyTorch:

```python
nn.Linear(dim_in, dim_out)
```

Keras:

```python
layers.Dense(dim_out)
```

(2 demonstrations omitted)

PyTorch:

```python
F.log_softmax(logits, dim=-1)
```

Keras:

```python
tf.nn.log_softmax(logits, axis=-1)
```

PyTorch:

```python
nn.Conv2d(64, 128, 3)
```

Keras:
```
```
Conv2D(128, 3)
```

```
You are an expert in deep learning.
Transpile PyTorch to Keras:

PyTorch:

```python
F.log_softmax
```

Keras:

```python
tf.nn.log_softmax
```

PyTorch:

```python
nn.MaxPool2d
stride
```

Keras:

```python
layers.MaxPooling2D
strides
```

(7 demonstrations omitted)

PyTorch:

```python
F.relu
```

Keras:

```python
tf.nn.relu
```

PyTorch:

```python
nn.Conv2d
out_channels
```

Keras:
```
```
Conv2D
filters
```

Table 9: **Results of CSLS.** By default, ADELT computes similarity scores using *cosine similarity* to generate an API keyword dictionary. In this experiment, we replace cosine similarity with *inner product* or *cosine-CSLS-5* to compare different similarity measures. There are two numbers in each table cell: the first one is for transpiling PyTorch to PyTorch, and the second one is for transpiling Keras to PyTorch.

| | Keyword | | | Source-to-Source | |
| --- | --- | --- | --- | --- | --- |
| | P@1 | P@5 | MRR | BLEU | F1 |
| ADELT (Small) | 82.92 90.00 | 91.67 **97.73** | 86.97 94.04 | 93.83 **92.13** | 80.67 80.90 |
| ADELT (Base) | **87.08** 90.00 | 91.67 **97.73** | 89.67 93.96 | **95.32** 91.29 | **85.72 82.01** |
| Inner Product (Small) | 81.25 79.55 | 91.67 90.00 | 86.34 85.38 | 93.24 88.49 | 78.67 77.08 |
| Inner Product (Base) | 85.42 **93.18** | 91.67 **97.73** | 88.84 **95.71** | 94.38 91.75 | 82.17 81.46 |
| cos-CSLS-5 (Small) | 84.17 83.18 | **97.92** 93.64 | 89.89 89.12 | 94.24 90.43 | 83.17 76.60 |
| cos-CSLS-5 (Base) | **87.08** 89.55 | 97.50 **97.73** | **90.63** 93.75 | 95.20 90.27 | 85.39 76.18 |

Table 10: **Full results with 95% confidence intervals**. For each experiment, we run five experiments with different random seeds. Each cell has two intervals: the first one is for transpiling PyTorch to Keras, and the second one is for transpiling Keras to PyTorch. Each interval is the 95% confidence interval according to the Student's t-Test, where we assume that the result of the five experiments follows a normal distribution.

| | Keyword | | Source-to-Source | |
| --- | --- | --- | --- | --- |
| | P@1 | | F1 | |
| *LM few shot* | | | | |
| GPT-3 (Brown et al., 2020) | $35.4 \pm 6.1$ | $39.1 \pm 4.2$ | $26.6 \pm 5.1$ | $32.1 \pm 6.7$ |
| Codex (Chen et al., 2021) | $67.5 \pm 8.3$ | $79.1 \pm 7.8$ | $59.9 \pm 2.7$ | $67.1 \pm 2.2$ |
| GPT-4 | $74.5 \pm 5.8$ | $83.2 \pm 3.5$ | $67.7 \pm 2.6$ | $74.9 \pm 1.7$ |
| *ADELT* | | | | |
| ADELT (Small) | $82.9 \pm 1.2$ | $90.0 \pm 1.6$ | $79.0 \pm 2.2$ | $76.7 \pm 1.5$ |
| ADELT (Base) | $87.1 \pm 1.2$ | $90.0 \pm 2.5$ | $83.4 \pm 0.8$ | $82.0 \pm 2.2$ |
| w/o PyBERT (Small) | $52.1 \pm 2.6$ | $63.6 \pm 4.5$ | $37.2 \pm 8.2$ | $43.0 \pm 4.5$ |
| w/o PyBERT (Base) | $45.0 \pm 3.9$ | $54.6 \pm 5.3$ | $33.0 \pm 6.5$ | $36.3 \pm 3.2$ |
| w/o Adv Loss (Small) | $80.4 \pm 1.4$ | $88.6 \pm 2.0$ | $65.8 \pm 2.0$ | $73.6 \pm 1.8$ |
| w/o Adv Loss (Base) | $86.3 \pm 1.4$ | $90.5 \pm 2.4$ | $78.2 \pm 2.1$ | $72.3 \pm 3.6$ |
| Dot product (Small) | $82.9 \pm 4.5$ | $90.0 \pm 7.2$ | $74.6 \pm 2.7$ | $73.2 \pm 3.4$ |
| Dot product (Base) | $87.1 \pm 1.2$ | $90.0 \pm 2.0$ | $80.2 \pm 0.7$ | $78.8 \pm 0.8$ |

Table 11: **A synthetic example of convolution layer from the evaluation dataset of the Keras-PyTorch transpilation task.** We show the Keras code, ground truth PyTorch code, and the outputs from Codex, ADELT, and ADELT +. ✓: the output is the same or equivalent to the ground truth. ✓: the output contains an equivalent of the ground truth, but it also contains incorrect extra code. ✗: the output is incorrect.

| | |
|---|---|
| Source | ```
in_dim = 64
out_dim = 128
layers.Conv2D(filters=out_dim,
  kernel_size=3,
  activation="relu")
``` |
| Truth | ```
in_dim = 64
out_dim = 128
nn.Conv2d(in_dim, out_dim, 3)
nn.ReLU()
``` |
| Codex ✗ | ```
in_dim = 64
out_dim = 128
nn.Conv2d(in_dim, out_dim, 3)
``` |
| ADELT ✗ | ```
in_dim = 64
out_dim = 128
nn.Linear(in_features=in_dim,
  out_features=out_dim,
  kernel_size=3)
``` |
| ADELT + ✓ | ```
in_dim = 64
out_dim = 128
nn.Conv2d(in_channels=in_dim,
  out_channels=out_dim,
  kernel_size=3)
nn.ReLU()
``` |

Table 12: Example of transpiling from PyTorch in Python 2 to Keras in Python 3.

*Canonicalized Source Program in Python 2*

```python
import torch.nn as nn
dense = nn.Linear(in_features=dim_in / 2, out_features=dim_out / 2, bias=False)
```

*Code Skeleton*

```python
import torch.nn as nn
print dim_in, dim_out
dense = PLACEHOLDER_1(PLACEHOLDER_2=dim_in / 2, PLACEHOLDER_3=dim_out / 2, PLACEHOLDER_4=False)
```

*Codex Input*

```python
# Translate from PyTorch in Python2 to Keras in Python3

# PyTorch in Python2
PLACEHOLDER_1

# Keras in Python3
PLACEHOLDER_1

# PyTorch in Python2
PLACEHOLDER_2

# Keras in Python3
PLACEHOLDER_2

# PyTorch in Python2
PLACEHOLDER_3

# Keras in Python3
PLACEHOLDER_3

# PyTorch in Python2
import torch.nn as nn
class Model(nn.Module):
    def __init__(self):
        super().__init__()
        print "Building Model"
        self.layer1 = PLACEHOLDER_1(PLACEHOLDER_2=16 / 2, PLACEHOLDER_3=32, PLACEHOLDER_4=3)
        self.layer2 = PLACEHOLDER_5()

    def forward(self, x):
        x = self.layer1(PLACEHOLDER_6=x)
        x = self.layer2(PLACEHOLDER_7=x)
        return x

# Keras in Python3
import tensorflow.keras.layers as layers
class Model(layers.Layer):
    def __init__(self):
        super().__init__()
        print("Building Model")
        self.layer1 = PLACEHOLDER_1(PLACEHOLDER_2=16 // 2, PLACEHOLDER_3=32, PLACEHOLDER_4=3)
        self.layer2 = PLACEHOLDER_5()

    def call(self, x):
        x = self.layer1(PLACEHOLDER_6=x)
        x = self.layer2(PLACEHOLDER_7=x)
        return x

# PyTorch in Python2
import torch.nn as nn
print dim_in, dim_out
dense = PLACEHOLDER_1(PLACEHOLDER_2=dim_in / 2, PLACEHOLDER_3=dim_out / 2, PLACEHOLDER_4=False)

# Keras in Python3
```

*Codex Output*

```python
import tensorflow.keras.layers as layers
print(dim_in, dim_out)
dense = PLACEHOLDER_1(PLACEHOLDER_2=dim_in // 2, PLACEHOLDER_3=dim_out // 2, PLACEHOLDER_4=False)
```

*Target Program in Python 3*

```python
import tensorflow.keras.layers as layers
print(dim_in, dim_out)
dense = layers.Dense(units=dim_out // 2, use_bias=False)
```

