# OpenReview forum: "ADELT: Transpilation Between Deep Learning Frameworks"
_ICLR.cc/2024/Conference — Submitted to ICLR 2024_

### Official Review · Reviewer_hxrU · 2023-10-31

**Soundness:** 4 excellent
**Presentation:** 3 good
**Contribution:** 4 excellent
**Rating:** 6
**Confidence:** 4

**Summary:**

This paper introduce a novel source-to-source transpilation framework between deep learning frameworks, named ADELT.
The core innovation is to decouple the process into code skeleton transpilation and API keyword mapping, so more structural information and domain information could be injected through prompting and adversarial training.
The proposed method outperforms the SOTA method by a good margin on PyTorch-Keras and PyTorch-MXNet transpilation pairs.

**Strengths:**

I think the proposed ADELT is solid and innovative.
I like the idea of decoupling an end-to-end generation process into separate parts, which allows more context/structural information to be incorporated into the process. But it's often hard to achieve because of the complexity of different tasks. For the transpilation task, I think the author find a good combination of the code skeleton and the API keywords mapping. And the experimental results show a significant improvement over the SOTA system, which is quite convincing.
Overall I think it's a solid paper, well-written and easy to understand.

**Weaknesses:**

One suggestion is that in the code skeleton process, all the API keywords are replaced by placeholders (namely PLACEHOLDER_1,2 ...).
I wonder if it will be beneficial to add more context information into the placeholders, given them more concrete meanings.
For example in machine translation, the name entity is often replaced as <NUM>, <TIME>, <NAME> etc, which will give more context information in the downstream translation process.   So maybe the same idea could be applied in transpilation.

**Questions:**

As the transpilation is decoupled into two sub-parts:  the code skeleton and the API keywords mapping.
Is it possible to measure the precision of each process and how do they effect the final performance ?

---

> ### Author Response · Authors · 2023-11-23
> **Response to Reviewer hxrU**
>
> Thank you for your encouraging comments! We address your concerns below:
>
> **W1. Why not add more context information into the placeholders**
>
> We appreciate your insightful recommendations. Rather than using generic placeholders like `[PLACEHOLDER_1]`, `[PLACEHOLDER_2]`, we agree that using more specific tags such as `[FUNC_1]`, `[ARG_1]` could provide valuable context. However, given that our current approach has achieved 100% accuracy for code skeleton transpilation in our benchmark evaluations, this modification might be not necessary for this given task. That said, your suggestion will be useful for more challenging tasks, and will be considered in future work. Thank you again for your valuable input.
>
> **Q1. Is it possible to measure the precision of each process (the code skeleton and the API keywords mapping) and how do they effect the final performance?**
>
> We appreciate your insightful question regarding the precision of our transpilation process's sub-parts and their subsequent effect on ADELT's overall performance.
>
> For skeletal code transpilation, Section 3.4 describes its accuracy of 100% on the benchmark, because skeletal code transpilation is designed to be simple and straightforward.
>
> For dictionary lookup, the performance is reported in Table 3 (ablation studies), where we show metrics such as Precision@1, Precision@5, and MRR. The dictionary of ADELT has Precision@1 of 87.1% and 90.0% for both transpilation directions respectively.

---

### Official Review · Reviewer_DL7Y · 2023-10-31

**Soundness:** 1 poor
**Presentation:** 2 fair
**Contribution:** 2 fair
**Rating:** 3
**Confidence:** 4

**Summary:**

This paper presents a method translate short code snippets from one deep learning framework to another. The authors create a mapping of keywords from one framework to another, using a new BERT-like model pre-trained on Python code. This model is augmented with a generator and discriminator, and finetuned using unsupervised data to maximize cosine similarity, similar to dense retrieval methods. The authors then pipeline prompting LLMs to translate the code, and then use their mapping to replace the keywords from the source framework to the target. The proposed method outperforms prompting LLMs, unsupervised NMT approaches, and simple heuristic approaches on a small evaluation set of 50 samples of around 10 lines each.

**Strengths:**

1. A BERT model trained on Python code, which could have interesting applications - for example for code refactoring, static analysis, etc.
1. Using retrieval cosine similarity to map API keywords from one framework to another, which is learnt using fully unsupervised and unaligned data.

**Weaknesses:**

1. Very small evaluation dataset size of 50 samples. This makes any evaluation or comparisons made on this data less robust, with 95% deviations of upto +- 8.8 points.
1. The evaluation dataset construction process itself introduces biases -
    1. By selecting only pairs with high BLEU score, this makes the dataset somewhat simpler than what a more uniform sampling might have yielded (say with a manually created dataset, with humans transpiling the code). This "simpler" dataset then in turn hides how the proposed method may compare to others (such as prompting LLMs) in more complicated instances.
    1. The BLEU score filtering also biases the dataset towards shorter snippets with $\approx10$ lines of code - This restricts the scope of this method, to real world scenarios with very small code snippets to translate, and again makes realistic comparisons difficult.
1. The pre-training dataset used may itself have been on interest, but perhaps will not be feasible to release due to copyright concerns.

**Questions:**

1. For table 9, could the authors provide GPT4's 95% intervals, specifically for F1? (corresponding to scores in Table 1)

Presentation suggestion -
1. 2 decimal digits (in Table 1, etc) make it hard to read the table to quickly compare values. 1 decimal digit will be easier/faster to understand. However, given the 95% confidence intervals are so large, any decimal digit is meaningless - Perhaps no decimal digits should be shown.
1. Datasets in A.1 are reported in GBs, but often a significant fraction of ipynb notebook files is binary data in output cells - a better metric to report here would be either tokens, or dataset size without the binary parts of ipynb.

---

> ### Author Response · Authors · 2023-11-23
> **Response to Reviewer DL7Y (1/2)**
>
> **W1. Very small evaluation dataset size of 50 samples. 95% confidence interval of upto +- 8.8 points.**
>
> Thank you for your insightful feedback regarding the evaluation dataset's size. Recognizing the importance of a robust and reliable assessment of ADELT's performance, we have expanded the evaluation dataset to 100 examples. The additional 50 examples were curated by identifying PyTorch modules sampled from GitHub repositories with over 1,000 stars. Human experts subsequently translated these into the target frameworks, and you can find the selection process detailed in Appendix A.4 of the revised manuscript.
>
> Upon re-evaluating the PyTorch-Keras transpilation with the augmented benchmark, ADELT maintains a substantial performance lead, achieving an improved exact match score increase of 17.4 percentage points over the baseline models. Updated results have been incorporated into the relevant sections of our manuscript. We believe that this improvement in our evaluation substantiates the reliability of our results.
>
> The size of the revised dataset is comparable to the standard in domain-specific benchmarks: the general-purpose code generation dataset OpenAI HumanEval contains 164 examples, and most program synthesis benchmarks comprise less than 100 examples, like Casper (37 examples) [1], C2TACO (71 examples) [2], TF-Coder (70 examples) [3]. While we were able to only double the size of our evaluation dataset due to the limited time over the rebuttal period, we will expand the dataset further post-rebuttal.
>
> Regarding statistical significance, 95% deviation of 8.88 points was observed from F1 score of GPT-3 (35.62 ± 8.88), which is statistically significantly inferior to ADELT originally reported (82.01 ± 3.08).
>
> We have introduced a full results table with error bars for GPT-4 and other models on our new 100-example benchmark (also shown in Table 10 of the revised manuscript):
>
> |   |    **Keyword**   | **(Precision@1)** | **Source-to-Source** | **(F1 Score)**   |
> |-|:-|:-|:-|:-|
> | | PyTorch-to-Keras | Keras-to-PyTorch  |   PyTorch-to-Keras   | Keras-to-PyTorch |
> | GPT-3 |     35.4±6.1     | 39.1±4.2          | 26.6±5.1             | 32.1±6.7         |
> | Codex  |     67.5±8.3     | 79.1±7.8          | 59.9±2.7             | 67.1±2.2         |
> | GPT-4 |     74.5±5.8     | 83.2±3.5          | 67.7±2.6             | 74.9±1.7         |
> | ADELT (Small) |     82.9±1.2     | **90.0±1.6**      | 79.0±2.2             | 76.7±1.5         |
> | ADELT (Base) |   **87.1±1.2**   | **90.0±2.5**      | **83.4±0.8**         | **82.0±2.2**     |
>
> These results illustrate that ADELT (Base) surpasses GPT-4 in all benchmarks with a statistically significant difference. We hope that this larger dataset and comprehensive re-evaluation address your concerns and bring further rigor to our work.
>
> **W2. The evaluation dataset construction process itself introduces biases. By selecting only pairs with high BLEU score**
>
> Thank you for your astute observations concerning potential biases caused by our initial data collection method. You are correct that our initial reliance on BLEU score heuristics may have inadvertently skewed our dataset towards simpler examples.
>
> However, it's important to clarify that the BLEU score was used as an initial guide to locate repositories potentially containing parallel examples of code across deep learning frameworks, rather than as a strict selection criterion.
>
> In response to your suggestion, we have doubled our evaluation dataset with more challenging, realistic examples sourced from PyTorch repositories, which were then manually transpiled by human experts, as we discussed in our response to weakness 1.
>
> **W3. The pre-training dataset perhaps will not be feasible to release due to copyright concerns.**
>
> Thank you for your comment regarding the dataset used in our study. We acknowledge your concern about potential copyright issues related to parts of the data such as StackOverflow. To address this, while we have added these data in the supplementary material, we will release the script used to gather and process such data upon publication of the paper.
>
> [1] Leveraging Parallel Data Processing Frameworks with Verified Lifting, https://arxiv.org/abs/1611.07623
>
> [2] C2TACO: Lifting Tensor Code to TACO, https://dl.acm.org/doi/abs/10.1145/3624007.3624053
>
> [3] TF-Coder: Program Synthesis for Tensor Manipulations, https://arxiv.org/abs/2003.09040

---

> ### Author Response · Authors · 2023-11-23
> **Response to Reviewer DL7Y (2/2)**
>
> **Q1. For table 9, could the authors provide GPT4's 95% intervals, specifically for F1?**
>
> We apologize for the oversight in our original submission. The results with 95% confidence intervals for GPT-4, have now been added to Table 10 of the revised manuscript. We also discussed those results in our response to weakness 1. Thank you for bringing this to our attention.
>
> **PS1. 2 decimal digits (in Table 1, etc) make it hard to read the table to quickly compare values. 1 decimal digit will be easier/faster to understand.**
>
> Thank you for your valuable suggestion. We appreciate your attention to clarity and have updated our tables accordingly, presenting all numbers rounded to 1 decimal digit.
>
> **PS2. Datasets in A.1 are reported in GBs, but often a significant fraction of ipynb notebook files is binary data in output cells**
>
> We appreciate your insight. In response, we have revised Appendix A.1 to reflect the size of clean Python code in all datasets, measured in gigabytes, thereby providing a more accurate sizing metric.

---

### Official Review · Reviewer_TwWZ · 2023-11-01

**Soundness:** 3 good
**Presentation:** 3 good
**Contribution:** 3 good
**Rating:** 6
**Confidence:** 4

**Summary:**

The paper introduces a novel approach called the Adversarial DEep Learning Transpiler (ADELT) for source-to-source transpilation between deep learning frameworks.

This work has two contributions, as follows:

1. ADELT is trained on an unlabeled web-crawled deep learning corpus, avoiding the need for hand-crafted rules or parallel data. It outperforms several related methods and significantly surpass the state-of-the-art large language model Codex, by 19.33 and 12.50 points, respectively.

2. The authors construct a PyTorch-Keras-MXNet corpus of deep learning code from various Internet sources, containing 19,796 PyTorch modules, 3,703 Keras layers/models, and 1,783 MXNet layers/models. An evaluation benchmark is then built to assess both the API keyword mapping algorithm and the overall source-to-source transpilation.

In summary, ADELT represents a significant advancement in the field of source-to-source transpilation for deep learning frameworks, offering improved performance without the need for labeled data and providing a valuable resource for the deep learning community by sharing code, corpus, and evaluation benchmarks openly.

**Strengths:**

1. Originality:

a. Decoupling of Transpilation Components: ADELT's approach to decoupling code skeleton transpilation and API keyword mapping is a unique and innovative contribution. This separation allows for more flexibility and adaptability in the transpilation process, and it sets ADELT apart from existing end-to-end methods.

b. Few-shot Prompting and Domain-Adversarial Training: The use of few-shot prompting on large language models for code transpilation and domain-adversarial training of contextual embeddings for API keyword mapping is a novel combination. It leverages the strengths of these techniques to enhance transpilation accuracy.

2. Quality:

a. Performance Improvement: ADELT demonstrates substantial improvements in transpilation accuracy compared to state-of-the-art transpilers. The increase in exact match scores for PyTorch-Keras and PyTorch-MXNet transpilation pairs is impressive, indicating the high quality of the proposed approach.

b. Data Utilization: ADELT's ability to achieve these results without relying on labeled data is a testament to its quality. Training on an unlabeled web-crawled deep learning corpus showcases the effectiveness of the approach in real-world scenarios.


Besides, the clear problem statement makes the problem of source-to-source transpilation accessible to a broad audience. The methodology, including the use of few-shot prompting, domain-adversarial training, and the construction of a corpus, is well documented, enhancing the clarity of the proposed approach. ADELT represents an important step forward in addressing the challenges of working with different deep learning frameworks.

**Weaknesses:**

1. Lack of Negative Results: The paper primarily focuses on the strengths and successes of ADELT. Including discussions on potential limitations, challenges, or cases where ADELT might not perform as well would make the paper more balanced and provide a more realistic perspective.

2. Generalization of the Approach: The paper primarily focuses on specific transpilation pairs like PyTorch-Keras and PyTorch-MXNet. To enhance the paper's significance, discussing the potential for ADELT's application to a broader range of deep learning frameworks and scenarios would be valuable.

**Questions:**

1. How does ADELT handle situations where different versions of deep learning frameworks require different expressions? Can you give more explanations? Such as,
for the old version of Python:
x = torch.autograd.Variable(torch.Tensor([1.0]), requires_grad=True)
for a newer version:
x = torch.tensor([1.0], requires_grad=True)

2. In the METHOD section, the authors mention that they "convert each API call into its canonical form". How exactly is this achieved? Can you provide a detailed explanation, or are there existing open-source tools for this purpose?

3. I don't quite understand what ${e_i^{(1)}}\_{i=1}^{m^{(1)}}$ and ${e_j^{(2)}}_{j=1}^{m^{(2)}}$ mean in the last paragraph of Section 2.3.

4. Instead of an end-to-end model, ADELT is actually a pipeline solution, consisting of multiple components, such as "Canonicalizion" "Extract API calls" "Code to Skeleton" and "Dictionary Lookup" and so on, right? Have the authors tested the accuracy of each component? They will ultimately impact the overall quality of the code skeleton transpilation.

---

> ### Author Response · Authors · 2023-11-23
> **Response to Reviewer TwWZ**
>
> **W1. Lack of Negative Results**
>
> Although our paper centered on ADELT's strengths, we have indeed discussed its limitations in detail.
>
> In Table 2, we provided a detailed case study illustrating instances where ADELT demonstrated less optimal performance. Notably, a case where the parameter `output_dim` was inaccurately mapped to `embeddings_initializer` due to an error in generating vocabulary mapping.
>
> Also, ADELT is unable to transpile `layers.Dense(out_dim, activation='relu')` into corresponding PyTorch API calls, `nn.Linear` and `nn.ReLU`. This is because the training corpus does not include enough examples that reflect such use cases in Keras. In Appendix A.9, we describe the potential for ADELT's improved modeling of such mappings in a scenario where we add synthetic data into the corpus.
>
> We value your perspective and will expand our discussion of both the pros and cons in our paper.
>
> **W2. Generalization of the Approach**
>
> We appreciate your insightful suggestion on generalizing the application of ADELT. While our current study focuses on transpilation between specific deep learning frameworks, ADELT's approach can be extended to various API ecosystems within the same programming language. We believe that ADELT’s approach to transpilation can also be applied to transpile between APIs from different domains, such as visualization (Matplotlib vs. Seaborn) or database access APIs (PyMongo vs SQLite). Your feedback has certainly contributed to outlining the future directions of our work.
>
> **Q1. ​​How does ADELT handle situations where different versions of deep learning frameworks require different expressions?**
>
> Thank you for your insightful question. ADELT addresses variability across framework versions by treating each version-specific expression as a unique entry. Both `torch.autograd.Variable` and `torch.tensor` in PyTorch are mapped to `tf.Tensor` in Keras. The reverse process, subsequently, interprets `tf.Tensor` from Keras as the more recent `torch.tensor` expression in PyTorch. This approach maintains consistency across different framework versions and ensures accurate transpilation.
>
> **Q2. In the METHOD section, the authors mention that they "convert each API call into its canonical form". How exactly is this achieved?**
>
> We convert each API call into its canonical form using Python's built-in `inspect` module, specifically `inspect.bind` (https://docs.python.org/3/library/inspect.html). This function interprets Python function calls internally and creates a mapping from positional and keyword arguments to parameters. We have further detailed this procedure in Section 2.1 of our paper.
>
> **Q3. What e_i and e_j mean in the last paragraph of Section 2.3.**
>
> Thank you for your question. In Section 2.3, $e_i$ and $e_j$ correspond to the unique embeddings of each API keyword. To elaborate, every API keyword, like `nn.Conv2d` in PyTorch, is assigned its unique embedding, like `nn.Linear` or `nn.Linear.in_features`. In Equation (2), we incorporate these embedding matrices as weight matrices for the output classification layer, a strategy that is commonly adopted in Transformer LMs where a shared parameter tensor is used between the word embedding and the output layer [1].
>
> **Q4. Have the authors tested the accuracy of each component, such as "Canonicalizion" "Extract API calls" "Code to Skeleton" and "Dictionary Lookup" and so on?**
>
> Yes, ADELT indeed follows a pipeline structure that includes “Canonicalization”, “API call extraction”, “Code to Skeleton”, “Skeletal Code Transpilation” and “Dictionary Lookup” among other steps.
>
> For “Canonicalization”, “API call extraction”, “Code to Skeleton”, our approach uses Python's internal library and does not involve machine learning, thereby not affecting transpilation accuracy.
>
> For “Skeletal Code Transpilation”, Section 3.4 describes its accuracy of 100% on the benchmarks. In fact, one of the key novelties of ADELT is the decoupling of the transpilation process into transpiling the skeletal code followed by dictionary lookup. Skeletal code transpilation is designed to be simple using any off-the-shelf LLM, while the dictionary used in the second step requires more sophisticated techniques to learn via adversarial learning.
>
> For “Dictionary Lookup”, the performance is reported in Table 3  (ablation studies), where we show metrics such as Precision@1, Precision@5, and MRR. The dictionary of ADELT has Precision@1 of 87.1% and 90.0% for both transpilation directions respectively.
>
> We hope that this clarifies the efficacy of ADELT's individual components.
>
> [1] Rethinking embedding coupling in pre-trained language models, https://arxiv.org/abs/2010.12821

---

### Official Review · Reviewer_MAQg · 2023-11-01

**Soundness:** 3 good
**Presentation:** 3 good
**Contribution:** 2 fair
**Rating:** 3
**Confidence:** 5

**Summary:**

The paper introduces Adversarial DEep Learning Transpiler (ADELT) for source-to-source transpilation between deep learning frameworks. It accomplishes this by transpiling the code's skeleton using a pretrained language model and mapping keywords through a keyword translation dictionary generated by domain-adversarial training process. The author also built an evaluation benchmark for PyTorch-Keras and PyTorch-MXNet transpilation.

**Strengths:**

Strengths:

1. The method proposed in this article seems to be effective. ADELT uniquely decouples the code skeleton transpilation and API keyword mapping, which allows for greater flexibility and adaptability in the transpilation process.

2. The authors establish a small evaluation dataset. The paper conducts comprehensive ablation studies.

**Weaknesses:**

1. The description in the Method section is somewhat challenging to understand. For instance, the example in Figure 1 is too simple, and it's not clear how the complete actions of step 2 and step 3 are reflected. Additionally, based on the description, it seems like steps 3, 4, and 5 are executed sequentially. However, in Figure 1, it appears that step 4 is executed in parallel with the steps 3,5, which is somewhat perplexing.

2. The size of the evaluation dataset is somewhat small, with only 50 samples, making it challenging to ensure the reliability of the experiments and comparisons.

3. Some details of the comparisons were not clearly explained, such as what prompts were used for GPT-4, as the choice of prompts can significantly impact the model's performance.

4. If I did not overlook, the article only provides an account of the construction process for the PyTorch-Keras benchmark in Appendix A.4, but, it does not cover the construction process for the PyTorch-MXNet benchmark.

5. The execution efficiency also needs some comparisons, such as runtime, as the speed of a pipeline is typically expected to be slower than a single end-to-end model. However, the paper does not quantify the magnitude of this difference.

**Questions:**

In section 2.4, what do you mean by “In dictionary generation, we do not allow callable names to be translated to callable names”? Can you provide a detailed explanation or an example to explain it?

---

> ### Author Response · Authors · 2023-11-23
> **Response to Reviewer MAQg (1/2)**
>
> **W1. Figure 1 is too simple. It appears that step 4 is executed in parallel with the steps 3, 5**
>
> Thank you for your insightful comments regarding the Method section of our paper. We apologize for any confusion caused by our initial presentation in Figure 1. We have revised the figure in the updated manuscript to better depict our approach. The updated Figure 1 now provides a more detailed illustration, clarifying how keyword substitutions from step 4 integrate with the other steps.
>
> Yes, steps 3 and 4 are executed concurrently, each building upon the output of step 2, while step 5 synthesizes the results from both steps 3 and 4.
>
> **W2. The size of the evaluation dataset is somewhat small, with only 50 samples**
>
> Thank you for your insightful feedback regarding the evaluation dataset's size. Recognizing the importance of a robust and reliable assessment of ADELT's performance, we have expanded the evaluation dataset to 100 examples. The additional 50 examples were curated by identifying PyTorch modules sampled from GitHub repositories with over 1,000 stars. Human experts subsequently translated these into the target frameworks, and you can find the selection process detailed in Appendix A.4 of the revised manuscript.
>
> Upon re-evaluating the PyTorch-Keras transpilation with the augmented benchmark, ADELT maintains a substantial performance lead, achieving an improved exact match score increase of 17.4 points over the baseline models. Updated results have been incorporated into the relevant sections of our manuscript. We believe that this improvement in our evaluation substantiates the reliability of our results.
>
> The size of the revised dataset is comparable to the standard in domain-specific transpilation benchmarks: the general-purpose code generation dataset OpenAI HumanEval contains 164 examples, and most program synthesis benchmarks comprise of less than 100 examples, like Casper (37 examples) [1], C2TACO (71 examples) [2], TF-Coder (70 examples) [3]. While we were able to only double the size of our evaluation dataset due to the limited time over the rebuttal period, we will expand the dataset further post-rebuttal.
>
> **W3. Some details of the comparisons were not clearly explained, such as what prompts were used for GPT-4**
>
> Thank you for your insightful comments. We apologize for missing the GPT-4 prompt details. To address this, we have included a comprehensive explanation of the prompts used for GPT-4 in Appendix A.6 and detailed the post-processing protocol in Table 8 of the revised manuscript. As we explain in the appendix, our experiments leverage Markdown-formatted prompts in GPT-4's chat mode, and then GPT-4 sometimes generates a mixture of natural language and code. After generation, we programmatically extract code from the first code block from GPT-4's responses as the finalized transpilation. Our prompts for GPT-4 are designed to mimic the straightforward and intuitive approach of a human user, as done in prior approaches in evaluating such models [4].
>
>
> **W4. The paper only provides the construction process for the PyTorch-Keras benchmark in Appendix A.4, but not cover the construction process for the PyTorch-MXNet benchmark**
>
> Thank you for pointing out this oversight. We have now detailed the construction method for the PyTorch-MXNet benchmark in the revised Appendix A.4. Briefly, human experts manually transpiled the PyTorch snippets from our PyTorch-Keras dataset into MXNet, ensuring consistency across frameworks. Your feedback is greatly appreciated for the improvement of our manuscript.
>
> **W5. The execution efficiency also needs some comparisons, such as runtime**
>
> We appreciate your concern on ADELT's computational efficiency. Efficiency is actually one of the strengths of ADELT.
>
> ADELT is composed of two main procedures: large language model (LLM) prompting for code transpilation and dictionary lookup for API keyword mapping. Using Codex for transpiling the code skeleton is as efficient as using Codex to do transpilation end-to-end. The API dictionary lookup process is highly efficient and only takes nanoseconds since it does not involve neural network inference for each example.
>
> To illustrate, ADELT outperforms GPT-4 in transpilation results, and its skeletal code transpilation component uses Codex, which is of a similar scale as GPT-3. Because GPT-3/Codex is significantly faster, ADELT is faster than GPT-4.
>
> [1] Leveraging Parallel Data Processing Frameworks with Verified Lifting, https://arxiv.org/abs/1611.07623
>
> [2] C2TACO: Lifting Tensor Code to TACO, https://dl.acm.org/doi/abs/10.1145/3624007.3624053
>
> [3] TF-Coder: Program Synthesis for Tensor Manipulations, https://arxiv.org/abs/2003.09040
>
> [4] GPT-4 Technical Report, https://arxiv.org/abs/2303.08774

---

> ### Author Response · Authors · 2023-11-23
> **Response to Reviewer MAQg (2/2)**
>
> **Q1. In section 2.4, what do you mean by “In dictionary generation, we do not allow callable names to be translated to callable names”?**
>
> We appreciate your keen observation. The mentioned statement was indeed a mistake. The sentence should be, "In dictionary generation, we do not allow callable names to be translated to **parameter** names". This mistake has been rectified in the revised manuscript. Thanks for pointing it out!

---

### Author Response · Authors · 2023-11-23
**General Response**

We sincerely thank all reviewers for their insightful and constructive feedback. We are grateful for the reviewers’ recognition of our work in various aspects: the novelty of our ideas (Rewiewer hxrU), the effectiveness of our method (Reviewer MAQg, TwWZ, hxrU), the comprehensiveness of our experiments, especially ablation studies (Reviewer MAQg, hxrU), the diversity of our dataset (Reviewer TwWZ), and our method's unsupervised nature (Reviewer DL7Y).

Following reviewers' suggestions, we implemented the following revisions, with changes marked in blue in our updated manuscript:

1. As suggested by Reviewer MAQg, we have refined the diagram in Figure 1 to improve clarity, clarifying how keyword substitutions from step 4 integrate with the other steps.

2. As suggested by Reviewer MAQg and DL7Y, we have doubled our evaluation benchmark to 100 examples. The additional 50 examples have been sourced from GitHub and transpiled manually by expert developers to the target frameworks. Details of this process can be found in the updated Appendix A.4. All experiments were run again and the results have been updated accordingly.

3. As suggested by Reviewer DL7Y, we simplified all tables to present numbers up to one decimal place for easier reading.

4. As suggested by Reviewer DL7Y, we rectified a typo in Section 2.4.

5. As suggested by Reviewer DL7Y, we adjusted Appendix A.1 to show the size of clean Python code across all datasets for a more precise measure, replacing the size of original ipynb files that may count the size of execution logs and binaries.

6. As suggested by Reviewer MAQg, we included a detailed description of prompts used for GPT-4 and the post-processing protocol in Appendix A.6 and Table 8.

7. As suggested by Reviewer DL7Y, we added 95% confidence interval results for GPT-4 to Table 10. ADELT (Base) surpasses GPT-4 in all benchmarks with a statistically significant difference.

We extend our sincere thanks once again to all reviewers. Your constructive feedback has been invaluable in guiding us to refine and enhance the quality of our paper.

In the following sections, we address the specific concerns and questions raised by each reviewer.

---

### Meta-Review · Area_Chair_CdFB · 2023-12-18

**Metareview:**

The authors present ADELT, a framework for source-to-source transpilation between deep learning frameworks by decoupling code skeleton transpilation and API keyword mapping. Reviewers recognized ADELT's performance improvements and contributions to the transpilation task. Nevertheless, the reviewers have concerns about the small evaluation dataset and the need for more comprehensive validation across different frameworks and real-world scenarios.

While the authors appreciate the positive feedback and have addressed concerns by increasing the evaluation dataset size and providing error margins for GPT-4, concerns remain regarding the dataset's representativeness and the broader applicability of the approach. Reviewer hxrU suggested further refinement to increase the contextual specificity of placeholders, a point which the authors acknowledged while deeming it unnecessary based on their current evaluation metrics. The authors are encouraged to continue refining their approach and to consider expanding the evaluation further in future work.

**Justification For Why Not Higher Score:**

The reviewers have concerns about the small evaluation dataset and the need for more comprehensive validation across different frameworks and real-world scenarios.

**Justification For Why Not Lower Score:**

N/A

---

### Decision · Program_Chairs · 2024-01-16

Reject